# Evolutionary Expectation Maximization for Generative Models with Binary Latents

## Abstract

We establish a theoretical link between evolutionary algorithms and variational parameter optimization of probabilistic generative models with binary hidden variables. While the novel approach is independent of the actual generative model, here we use two such models to investigate its applicability and scalability: a noisy-OR Bayes Net (as a standard example of binary data) and Binary Sparse Coding (as a model for continuous data). Learning of probabilistic generative models is first formulated as approximate maximum likelihood optimization using variational expectation maximization (EM). We choose truncated posteriors as variational distributions in which discrete latent states serve as variational parameters. In the variational E-step, the latent states are then optimized according to a tractable free-energy objective. Given a data point, we can show that evolutionary algorithms can be used for the variational optimization loop by (A) considering the bit-vectors of the latent states as genomes of individuals, and by (B) defining the fitness of the individuals as the (log) joint probabilities given by the used generative model. As a proof of concept, we apply the novel evolutionary EM approach to the optimization of the parameters of noisy-OR Bayes nets and binary sparse coding on artificial and real data (natural image patches). Using point mutations and single-point cross-over for the evolutionary algorithm, we find that scalable variational EM algorithms are obtained which efficiently improve the data likelihood. In general we believe that, with the link established here, standard as well as recent results in the field of evolutionary optimization can be leveraged to address the difficult problem of parameter optimization in generative models.

## 1 Introduction

Evolutionary algorithms (EA) have been introduced (e.g. Fogel et al., 1966; Rechenberg, 1965) as a technique for function optimization using methods inspired by biological evolutionary processes such as mutation, recombination, and selection. As such EAs are of interest as tools to solve Machine Learning problems, and they have been frequently applied to a number of tasks such as clustering (Pernkopf & Bouchaffra, 2005; Hruschka et al., 2009), reinforcement learning (Salimans et al., 2017), and hierarchical unsupervised (Myers et al., 1999) or deep supervised learning (e.g., Stanley & Miikkulainen 2002 and Suganuma et al. 2017; Real et al. 2017 for recent examples). In some of these tasks EAs have been investigated as alternatives to standard procedures (Hruschka et al., 2009), but most frequently EAs are used to solve specific sub-problems. For example, for classification with Deep Neural Networks (DNNs LeCun et al., 2015; Schmidhuber, 2015), EAs are frequently applied to solve the sub-problem of selecting the best DNN architectures for a given task (e.g. Stanley & Miikkulainen, 2002; Suganuma et al., 2017) or more generally to find the best hyper-parameters of a DNN (e.g. Loshchilov & Hutter, 2016; Real et al., 2017).

Inspired by these previous contributions, we here ask if EAs and learning algorithms can be linked more tightly. To address this question we make use of the theoretical framework of probabilistic generative models and expectation maximization (EM Dempster et al., 1977) approaches for parameter optimization. The probabilistic approach in combination with EM is appealing as it establishes a very general unifying framework able to encompass diverse algorithms from clustering and dimensionality reduction (Roweis, 1998; Tipping & Bishop, 1999) over feature learning and sparse coding (Olshausen & Field, 1997) to deep learning approaches (Patel et al., 2016). However, for most generative data models, EM is computationally intractable and requires approximations. Variational

EM is a very prominent such approximation and is continuously further developed to become more efficient, more accurate and more autonomously applicable. Variational EM seeks to approximately solve optimization problems of functions with potentially many local optima in potentially very high dimensional spaces. The key observation exploited in this study is that a variational EM algorithm can be formulated such that latent states serve as variational parameters. If the latent states are then considered as genomes of individuals, EAs emerge as a very natural choice for optimization in the variational loop of EM.

## 2 TRUNCATED VARIATIONAL EM

A probabilistic generative model stochastically generates data points $\vec{y}$ using a set of hidden (or latent) variables $\vec{s}$. The generative process can be formally expressed in the form of joint probability $p(\vec{s}, \vec{y} \mid \Theta)$, where $\Theta$ are the model parameters. Given a set of $N$ data points, $\vec{y}^{(1)}, \ldots, \vec{y}^{(N)} = \vec{y}^{(1:N)}$, learning seeks to change the parameters $\Theta$ so that the data generated by the generative model becomes as similar as possible to the $N$ real data points. One of the most popular approaches to achieve this goal is to seek maximum likelihood (ML) parameters $\Theta^*$, i.e., parameters that maximize the data log-likelihood for a given generative model:

$$L(\Theta) := \log(\mathcal{L}(\Theta)) = \sum_n \log \left( \sum_{\{\vec{s}\}} p\left(\vec{y}^n, \vec{s} \mid \Theta\right)\right) \tag{1}$$

To efficiently find (approximate) ML parameters we follow Saul & Jordan (1996); Neal & Hinton (1998); Jordan et al. (1999) who reformulated the problem in terms of a maximization of a lower bound of the log-likelihood, the *free energy* $\mathcal{F}(\vec{q}, \Theta)$. Free energies are given by

$$\mathcal{F}(q^{(1:N)}, \Theta) = \sum_{n=1}^{N} \left( \sum_{\{\vec{s}\}} q^{(n)}(\vec{s}) \, \log \left(p(\vec{s}, \vec{y}^{(n)} \mid \Theta)\right)\right) + \sum_{n=1}^{N} H(q^{(n)}(\vec{s})), \tag{2}$$

where $q^{(n)}(\vec{s})$ are variational distributions, and where $H(q)$ denotes the entropy of a distribution $q$. For the purposes of this study, we consider elementary generative models which are difficult to train because of exponentially large state spaces. These models serve well for illustrating the approach but we stress that any generative model which gives rise to a joint distribution $p(\vec{s}, \vec{y} \mid \Theta)$ can be trained with the approach discussed here as long as the latents $\vec{s}$ are binary.

In order to find approximate maximum likelihood solutions, distributions $q^{(n)}(\vec{s})$ are sought that approximate the intractable posterior distributions $p(\vec{s} \mid \vec{y}^{(n)}, \Theta)$ as well as possible, which results in the free-energy being as similar (or tight) as possible to the exact log-likelihood. At the same time variational distributions have to result in tractable parameter updates. Standard approaches include Gaussian variational distributions (e.g. Opper & Winther, 2005) or mean-field variational distributions (Jordan et al., 1999). If we denote the parameters of the variational distributions by $\Lambda$, then a variational EM algorithm consists of iteratively maximizing $\mathcal{F}(\Lambda, \Theta)$ w.r.t. $\Lambda$ in the variational E-step and w.r.t. $\Theta$ in the M-step. The M-step can hereby maintain the same functional form as for exact EM but the expectation values now have to be computed w.r.t. the variational distributions.

Instead of using parametric functions such as Gaussians or factored (mean-field) distributions, for our purposes we choose truncated variational distributions defined as a function of a finite set of states (Lücke & Eggert, 2010; Sheikh et al., 2014; Shelton et al., 2017). These states will later serve as populations of evolutionary algorithms. If we denote $\mathcal{K}^n$ a population of hidden states for a given data point $\vec{y}^{(n)}$, then variational distributions and their corresponding expectation values are given by (e.g. Lücke & Eggert, 2010; Sheikh et al., 2014):

$$q^n(\vec{s} \mid \mathcal{K}^n, \Theta) := \frac{p\left(\vec{s} \mid \vec{y}^n, \Theta\right)}{\sum\limits_{\vec{s}' \in \mathcal{K}^n} p\left(\vec{s}' \mid \vec{y}^n, \Theta\right)} \delta(\vec{s} \in \mathcal{K}^n), \quad \langle g(\vec{s}) \rangle_{q^n} = \frac{\sum\limits_{\vec{s} \in \mathcal{K}^n} p(\vec{s}, \vec{y}^n \mid \Theta) g(\vec{s})}{\sum\limits_{\vec{s}' \in \mathcal{K}^n} p(\vec{s}', \vec{y}^n \mid \Theta)} . \tag{3}$$

where $\delta(\vec{s} \in \mathcal{K}^n)$ is 1 if $\mathcal{K}^n$ contains the hidden state $\vec{s}$, zero otherwise. If the set $\mathcal{K}^n$ contains all states with significant posterior mass, then (3) approximates expectations w.r.t. full posteriors very well. By inserting truncated distributions as variational distribution of the free-energy (2), it can be

shown (Lücke, 2016) that the free-energy takes a very compact simplified form given by:

$$\mathcal{F}(\mathcal{K}, \Theta) = \sum_n \log \left( \sum_{\vec{s} \in \mathcal{K}^n} p(\vec{y}^n, \vec{s} \mid \Theta) \right), \text{ where } \mathcal{K} = (\mathcal{K}^1, \ldots, \mathcal{K}^N). \tag{4}$$

As the variational parameters of the variational distribution (3) are now given by populations of hidden states, a variational E-step now consists of finding for each data point $n$ the population $\mathcal{K}^n$ that maximizes $\sum_{\vec{s} \in \mathcal{K}^n} p(\vec{y}^n, \vec{s} \mid \Theta)$.

## 3 EVOLUTIONARY OPTIMIZATION

For the generative models considered here, each latent state $\vec{s}$ takes the form of a bit vector. Hence, each population $\mathcal{K}^n$ is a collection of bit vectors. Because of the specific form (4), the free-energy is increased in the variational E-step if and only if we replace and individual $\vec{s}$ in population $\mathcal{K}^{(n)}$ by a new individual $\vec{s}^{\text{new}}$ so far not in $\mathcal{K}^{(n)}$ such that:

$$p(\vec{s}^{\text{new}}, \vec{y}^n \mid \Theta) > p(\vec{s}, \vec{y}^n \mid \Theta). \tag{5}$$

More generally, this means that the free energy is maximized in the variational E-step if we find for each $n$ those $S$ individuals with the largest joints $p(\vec{s}, \vec{y}^n \mid \Theta)$, where $p(\vec{s}, \vec{y}^n \mid \Theta)$ is given by the respective generative model (compare Lücke, 2016; Forster & Lücke, 2017, for formal derivations).

Full maximization of the free-energy is often a computationally much harder problem than increasing the free-energy; and in practice an increase is usually sufficient to finally approximately maximize the likelihood. As we increase the free-energy by applying (5) we can choose any fitness function $F(\vec{s}; \vec{y}^n, \Theta)$ for an evolutionary optimization which fulfils the property:

$$F(\vec{s}^{\text{new}}; \vec{y}^n, \Theta) > F(\vec{s}; \vec{y}^n, \Theta) \quad \Leftrightarrow \quad p(\vec{s}^{\text{new}}, \vec{y}^n \mid \Theta) > p(\vec{s}, \vec{y}^n \mid \Theta). \tag{6}$$

Any mutations selected such that the fitness $F(\vec{s}; \vec{y}^n, \Theta)$ increases will result in provably increased free-energies. Together with M-step optimizations of model parameters, the resulting variational EM algorithm will monotonously increase the free-energy. The freedom in choosing a fitness function satisfying (6) leaves us free to pick a form that enables an efficient parent selection procedure. More concretely (while acknowledging that other choices are possible) we define the fitness $F(\vec{s}^{\text{new}}; \vec{y}^n, \Theta)$ to be:

$$F(\vec{s}) = F(\vec{s}; \vec{y}^n, \Theta) = \widetilde{logP}(\vec{s}; \vec{y}^n, \Theta) - 2 \min_s \left( \widetilde{logP}(\vec{s}; \vec{y}^n, \Theta) \right) \tag{7}$$

where $\widetilde{logP}$ is defined as the logarithm of the joint probability where summands that do not depend on the state $\vec{s}$ have been elided. $\widetilde{logP}$ is usually more efficiently computable than the joint probabilities and has better numerical stability, while being a monotonously increasing function of the joints when the data-point $\vec{y}^n$ is considered fixed. As we will want to sample states proportionally to their fitness, an offset is applied to $\widetilde{logP}$ to make sure $F$ always takes positive values. As previously mentioned, other choices of $F$ are possible as long as (6) holds. From now on we will drop the argument $\vec{y}^n$ or index $n$ (while keeping in mind that an optimization is performed for each data point $\vec{y}^n$).

Our applied EAs then seek to optimize $F(\vec{s})$ for a population of individual $\mathcal{K}$ (we also drop the index $n$ here). More concretely, given the current population $\mathcal{K}$ of unique individuals $\vec{s}$, the EA iteratively seeks a new set $\mathcal{K}'$ with higher overall fitness. For our models, $\vec{s}$ are bit-vectors of length $H$, and we usually require that populations $\mathcal{K}'$ and $\mathcal{K}$ to have the same size as is customary for truncated approximations (e.g. Lücke & Eggert, 2010; Shelton et al., 2017). Our example algorithm includes three common genetic operators, discussed in more detail below: parent selection, generation of children by single-point crossover and stochastic mutation of the children. We repeat this process over $N_g$ generations in which subsequent iterations use the output of previous iterations as input population.

**Parent Selection.** This step selects $N_p$ parents from the population $\mathcal{K}$. Ideally, the selection procedure should be balanced between exploitation of parents with high fitness (which will more likely produce children with high fitness) and exploration of mutations of poor performing parents (which might eventually produce children with high fitness while increasing population diversity). Diversity is crucial, as $\mathcal{K}$ is a set of unique individuals and therefore the improvement of the overall

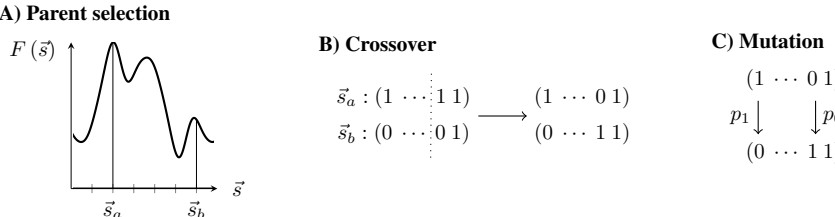

Figure 1: Components of the genetic algorithm.

fitness of the population depends on generating *different* children with high fitness. In our numerical experiments we explored both fitness-proportional selection of parents (a classic strategy in which the probability of an individual being selected as a parent is proportional to its fitness) and random uniform selection of parents.

**Crossover.** During the crossover step, random pairs of parents are selected; then each pair is assigned a number $c$ from 1 to $H - 1$ with uniform probability (this is the single crossover point); finally the parents swap the last $H - c$ bits to produce the offspring. We denote $N_c$ the number of children generated in this way. The crossover step can be skipped, making the EA more lightweight but decreasing variety in the offspring.

**Mutation.** Finally, each of the $N_c$ children undergoes one or more random bitflips to further increase offspring diversity. In our experiments we compare results of random uniform selection of the bits to flip with a more refined *sparsity-driven* bitflip algorithm. This latter bitflip schemes assigns to 0's and 1's different probabilities of being flipped in order to produce children with a sparsity compatible with the one learned by the model. In case the crossover step is skipped, a different bitflip mutation is performed on $N_c$ identical copies of each parent.

---

**Algorithm 1:** Evolutionary Expectation Maximization

choose initial model parameters $\Theta$ and initial sets $\mathcal{K}^{(n)}$
**repeat**
  **for** *each data-point $n$* **do**
    candidates = {}
    **for** $g = 0$ **to** $N_g$ **do**
      parents = select_parents
      children =
        mutation(crossover(parents))
      candidates = candidates $\cup$ children
    $\mathcal{K}^{(n)}$ = select_best($\mathcal{K}^{(n)} \cup$ candidates)
  update $\Theta$ using M-steps with (3) and $\mathcal{K}^{(n)}$
**until** $\mathcal{F}$ *has increased sufficiently*

---

A full run of the evolutionary algorithm therefore produces $N_g N_c N_p$ children (or new states $\vec{s}^*$). Finally we compute the union set of the original population $\mathcal{K}$ with all children and select the $S$ fittest individuals of the union as the new population $\mathcal{K}'$.

**The EEM Algorithm.** We now have all elements required to formulate a learning algorithm with EAs as its integral part. Alg. 1 summarizes the essential computational steps. Note that this E-step can be trivially parallelized over data-points. Finally, it is worth pointing out that algorithm 1, by construction, never decreases the free-energy.

## 4 THE GENERATIVE MODELS

We will use the EA formulated above as integral part of an unsupervised learning algorithm. The objective of the learning algorithm is the optimization of the log-likelihood 1. $D$ denotes the number of observed variables, $H$ the number of hidden units, and $N$ the number of data points.

**Noisy-OR.** The noisy-OR model is a highly non-linear bipartite data model with all-to-all connectivity among hidden and observable variables. All variables take binary values. The model assumes a Bernoulli prior for the latents, and active latents are then combined via the actual noisy-OR rule.

$$p(\vec{s} \mid \Theta) = \prod_h \pi_h^{s_h}(1 - \pi_h)^{1 - s_h} \qquad (8)$$

$$p(\vec{y} \mid \vec{s}, \Theta) = \prod_d N_d(\vec{s})^{y_d}(1 - N_d(\vec{s}))^{1 - y_d} \quad \text{where} \quad N_d(\vec{s}) := 1 - \prod_h (1 - W_{dh}s_h) \qquad (9)$$

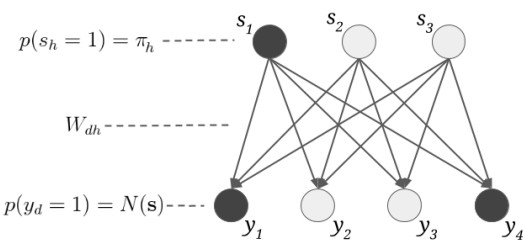

Figure 2: A small Noisy-OR model. Each observable $y_d$ is conditionally dependent on all $s_h$. The generative process first samples each $s_h$ from a Bernoulli distribution; then each $y_d$ is sampled from a Bernoulli distribution of parameter $N_d(\vec{s})$, generating a data-point.

In the context of the Noisy-OR model, $\Theta = \{\vec{\pi}, \vec{W}\}$, where $\vec{\pi}$ is the set of values $\pi_h \in [0, 1]$ representing the prior activation probabilities for the hidden variables $s_h$ and $\vec{W}$ is a $D \times H$ matrix of values $W_{dh} \in [0, 1]$ representing the probability that the latent $s_h$ activates the observable $y_d$.

Section A of the appendix contains the explicit forms of the free energies and the M-step update rules for noisy-OR.

**Binary Sparse Coding.** As a second model and one for continuous data, we consider Binary Sparse Coding (BSC; Henniges et al., 2010). BSC differs from standard Sparse Coding in its use of binary latent variables. The latents are assumed to follow a univariate Bernoulli distribution which uses the same activation probability for each hidden unit. The combination of the latents is described by a linear superposition rule. Given the latents, the observables are independently and identically drawn from a Gaussian distribution:

$$p\left(\vec{s} \,|\, \Theta\right) = \prod_{h=1}^{H} \pi^{s_h} \left(1 - \pi\right)^{1 - s_h}, \qquad p\left(\vec{y} \,|\, \vec{s}, \Theta\right) = \prod_{d=1}^{D} \mathcal{N}(y_d; \sum_{h=1}^{H} W_{dh} s_h, \sigma^2). \tag{10}$$

The parameters of the model are $\Theta = (\pi, W, \sigma^2)$, where $W$ is a $D \times H$ matrix whose columns contain the weights associated with each hidden unit $s_h$ and where $\sigma^2$ determines the variance of the Gaussian. M-step update rules for BSC can be derived in close-form by optimizing the free energy (2) wrt. all model parameters (compare, e.g., Henniges et al., 2010). We report the final expressions in appendix B.

## 5   NUMERICAL EXPERIMENTS

We describe numerical experiments performed to test the applicability and scalability of EEM. Throughout the section, the different evolutionary algorithms are named by indicating which parent selection procedure was used ("fitparents" for fitness-proportional selection, "randparents" for random uniform selection) and which bitflip algorithm ("sparseflips" or "randflips"). We add "cross" to the name of the EA when crossover was employed.

### 5.1   ARTIFICIAL DATA

First we investigate EMM using artificial data where the ground-truth components are known. We use the bars test as a standard setup for such purposes (Földiak, 1990; Hoyer, 2003; Lücke & Sahani, 2008). In the standard setup, $H^{\text{gen}}/2$ non-overlapping vertical and $H^{\text{gen}}/2$ non-overlapping horizontal bars act as components on $D = H^{\text{gen}} \times H^{\text{gen}}$ pixel images. $N$ images are then generated by first selecting each bar with probability $\pi^{\text{gen}}$. The bars are then superimposed according to the noisy-OR model (non-linear superposition) or according to the BSC model. In the case of BSC Gaussian noise is then added.

**Noisy-OR.** Let us start with the standard bars test which uses a non-linear superposition (Földiak, 1990) of 16 different bars (Spratling, 1999; Lücke & Sahani, 2008), and a standard average crowd-edness of two bars per images ($\pi^{\text{gen}} = \frac{2}{H^{\text{gen}}}$). We apply EEM for noisy-OR using different configurations of the EA. We use $H = 16$ generative fields. As a performance metric we here employ *reliability* (compare, e.g., Spratling, 1999; Lücke & Sahani, 2008), i.e., the fraction of runs whose learned free energies are above a certain minimum threshold and which learn the full dictionary of bars as well as the correct values for the prior probabilities $\pi$.

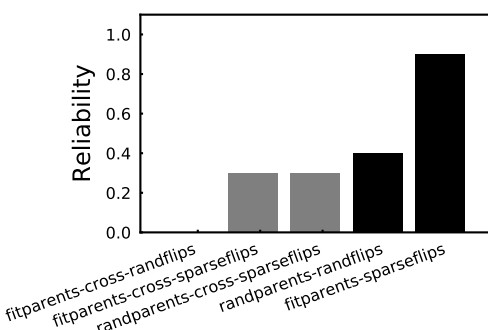

Figure 3: Reliability for the listed EAs over 10 runs of EEM for noisy-OR on 8x8 bars images. In this figure, black bars indicate both priors and bars were recovered correctly, grey bars indicate bars were recovered but not priors. For all runs $H = 16$, $N = 10^4$, $N_g = 2$, $N_p = 8$, $N_c = 7$, $S = 120$. Each run performed 100 iterations.

Figure 3 shows reliabilities over 10 different runs for each of the EAs. On 8x8 images the more exploitative nature of "fitparents-sparseflips" is advantageous over the simpler and more explorative "randparents-randflips". Note that this is not necessarily true for lower dimensionalities or otherwise easier-to-explore state spaces, in which also a naive random search might quickly find high-fitness individuals. In this test the addition of crossover reduces the probability of finding all bars and leads to an overestimation of the crowdedness $\pi H$.

After the initial verification on a standard bars test, we now make the component extraction problem more difficult by increasing overlap among the bars. A highly non-linear generative model such as noisy-OR is a good candidate to model occlusion effects in images. Figure 4 shows the results of training noisy-OR with EEM on a bars data-set in which the latent causes have sensible overlaps. The test parameters were chosen to be equal to those in (Lücke & Sahani, 2008, Fig. 9). After applying EEM with noisy-OR ($H = 32$) to $N = 400$ images with 16 strongly overlapping bars, we observed that all $H^{\mathrm{gen}} = 16$ bars were recovered in 13 of 25 runs, which is competitive especially when keeping in mind that no additional assumptions (e.g., compared to other models applied to this test) are used by EEM for noisy-OR.

Sample data-points   Learned weights

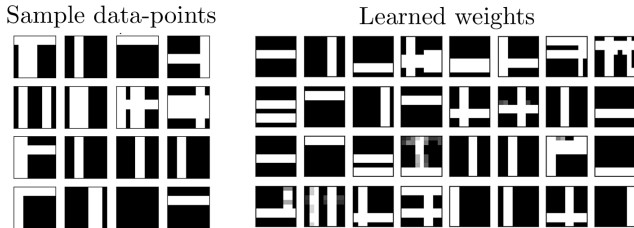

Figure 4: Sample input (left) and learned generative fields (right) for a run on overlapping bars. Out of 25 runs, 13 recovered all 16 ground-truth generative components (14.92 recovered bars in average, median 16). As $H = 32$, the extra generative fields are used to explain common overlaps and noise.

**BSC.** Like for the non-linear generative model, we first evaluate EEM for the linear BSC model on a bars test. For BSC, the bars are superimposed linearly (Henniges et al., 2010), which makes the problem easier. As a consequence, standard bars test were solved with very high reliability using EEM for BSC even if merely random bitflips were used for the EA. In order to make the task more challenging, we therefore (A) increased the dimensionality of the data to $D = 10 \times 10$ bars images, (B) increased the number of components to $H^{\mathrm{gen}} = 20$, and (C) increased the average number of bars per data point from two (the standard setting) to five. We employed $N = 5,000$ training data points and tested the same five different configurations of the EA as were evaluated for noisy-OR. We set the number of hidden units to $H = H^{\mathrm{gen}} = 20$ and used $S = 120$ variational states. Per data point and per iteration, in total 112 new states ($N_p = 8$, $N_c = 7$, $N_g = 2$) were sampled to vary $\mathcal{K}^n$. Per configuration of the EA, we performed 20 independent runs, each with 300 iterations. The results of the experiment are depicted in Fig. 5. We observe that a basic approach such as random uniform selection of parents and random uniform bitflips for the EA works well. However, more sophisticated EAs improve performance. For instance, combining bitflips with crossover and selecting parents proportionally to their fitness shows to be very benefical. The results also show that sparseness-driven bitflips lead generally to very poor performance, even if crossover or fitness-

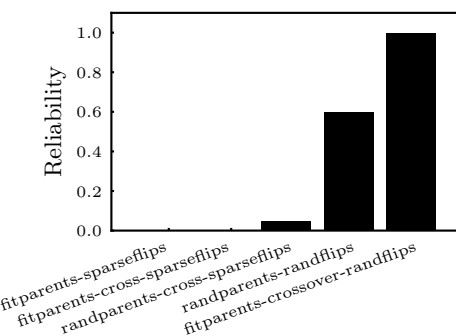

Figure 5: Reliability for the listed EAs over 20 runs of EEM for BSC on 10x10 bars images.

proportional selection of the parents is included. This effect may be explained with the initialization of $\mathcal{K}^n$. The initial states are drawn from a Bernoulli distribution with parameter $\frac{1}{H}$ which makes it more difficult for sparseness-driven EAs to explore and find solutions with higher crowdedness. Fig. 8 in appendix C depicts the averaged free energy values for this experiment.

## 5.2 NATURAL IMAGE PATCHES

Next, we verify the approach on natural data. We use patches of natural images, which are known to have a multi-component structure, which are well investigated, and for which typically models with high-dimensional latent spaces are applied. The image patches used are extracted from the van Hateren image database (van Hateren & van der Schaaf, 1998).

**Noisy-OR.** First we consider raw images patches, i.e., images without substantial pre-processing which directly reflect light intensities. Such image patches were generated by extracting random square subsections of a single 255x255 image of overlapping grass wires (part of image 2338 of the database). We removed the brightest 1% pixels from the data-set, scaled each data-point to have gray-scale values in the range $[0, 1]$ and then created data points with binary entries by repeatedly choosing a random gray-scale image and sampling binary pixels from a Bernoulli distribution with parameter equal to the gray-scale value of the original pixel (cfr. figure 6). Note that components in such light-intensity images can be expected to superimpose non-linearly because of occlusion, which motivates the application of a non-linear generative model such as noisy-OR. We employ the "fitparents-sparseflips" evolutionary algorithm that was shown to perform best on artificial data (3). Parameters were $H = 100$, $S = 120$, $N_g = 2$, $N_p = 8$, $N_c = 7$. Figure 6 shows the generative fields learned over 200 iterations. EEM allows learning of generative fields resembling curved edges, in line with expectations and with the results obtained in (Lücke & Sahani, 2008).

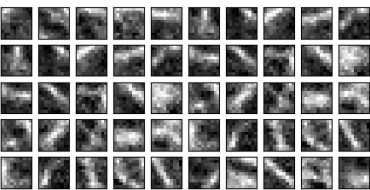

Figure 6: 50 generative fields learned by applying EEM ("fitparents-sparseflips") for noisy-OR to natural image patches. See Appendix F for a run at $H = 200$.

**BSC.** Finally, we consider pre-processed image patches using common whitening approaches as they are customary for sparse coding approaches (Olshausen & Field, 1997). We use $N = 100,000$ patches of size $D = 16 \times 16$, randomly picked from the whole data set. The highest 2% of the amplitudes were clamped to compensate for light reflections and patches without significant structure were excluded for learning. ZCA whitening (Bell & Sejnowski, 1997) was applied retaining 95% of the variance (we used the procedure of a recent paper Exarchakis & Lücke, 2017). We trained the BSC model for 4,000 iterations using the "fitparents-cross-sparseflips" EA and employing $H = 300$ hidden units and $S = 200$ variational states. Per data point and per iteration, in total 360 new states ($N_p = 10$, $N_c = 9$, $N_g = 4$) were sampled to vary $\mathcal{K}^n$. The results of the experiment are depicted in Fig. 7. The obtained generative fields primarily take the form of Gabor functions with different locations, orientations, phase, and spatial frequencies. This is a typical outcome of sparse coding being applied to images. On average more than five units were activated per data point showing that

the learned code makes use of the generative model's multiple causes structure. The generative fields converged faster than prior and noise parameters (similar effects are known from probabilistic PCA for the variance parameter). The finit slope of the free-energy after 4000 iterations is presumably due to these parameters still changing slowly.

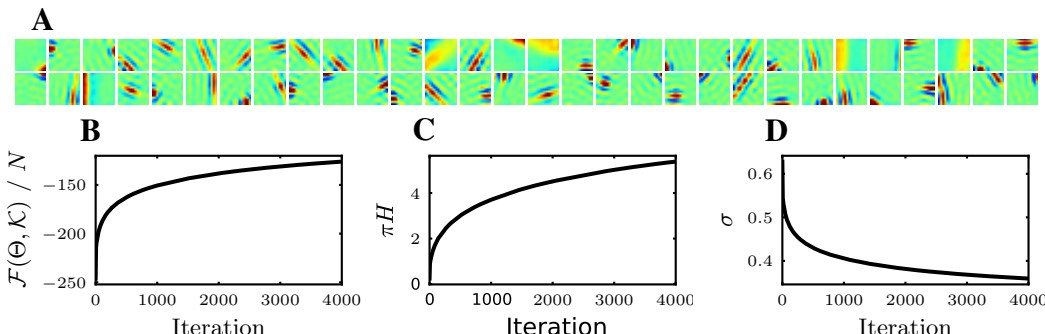

Figure 7: Results on training the BSC model on natural images using the "fitparents-cross-sparseflips" EA. **A** 60 of the 300 generative fields obtained through training (see Appendix for all fields). **B** Evolution of the free energy per data point over iterations. **C** Evolution of the expected number of active hidden units per data point over iterations. **D** Evolution of the standard deviation over iterations.

## 6 DISCUSSION

The training of generative models is a very intensively studied branch of Machine Learning. If EM is applied for training, most non-elementary models require approximations. For this reason, sophisticated and mathematically grounded approaches such as sampling or variational EM have been developed in order to derive sufficiently precise and efficient learning algorithms.

Evolutionary algorithms (EAs) have also been applied in conjunction with EM. Pernkopf & Bouchaffra (2005), for instance, have used EAs for clustering with Gaussian mixture models (GMMs). However, the GMM parameters are updated by their approach relatively convention- ally using EM, while EAs are used to select the best GMM models for the clustering problem (using a min. description length criterion). Such a use of EAs is similar to DNN optimization where EAs optimize DNN hyperparameters in an outer optimization loop (Stanley & Miikkulainen, 2002; Loshchilov & Hutter, 2016; Real et al., 2017; Suganuma et al., 2017, etc), while the DNNs them- selves are optimized using standard error-minimization algorithms. Still other approaches have used EAs to directly optimize, e.g., a clustering objective. But in these cases EAs *replace* EM approaches for optimization (compare Hruschka et al., 2009). In contrast to all such previous applications, we have here shown that EAs and EM can be combined directly and intimately: Alg. 1 defines EAs as an integral part of EM, and as such EAs address the key optimization problem arising in the training of generative models.

We see the main contribution of our study in the establishment of this close theoretical link between EAs and EM. This novel link will make it possible to leverage an extensive body of knowledge and experience from the community of evolutionary approaches for learning algorithms. Our numerical experiments are a proof of concept which shows that EAs are indeed able to train generative models with large hidden spaces and local optima. For this purpose we used very basic EAs with elementary selection, mutation, cross-over operators.

EAs more specialized to the specific optimization problems arising in the training of generative models have great potentials in future improvements of accuracy and scalability, we believe. In our experiments, we have only just started to exploit the abilities of EAs for learning algorithms. Still, our results represent, to the knowledge of the authors, the first examples of noisy-OR or sparse coding models trained with EAs (although both models have been studied very extensively before). Most importantly, we have pointed out a novel mathematically grounded way how EAs can be used for generative models with binary latents in general. The approach here established is, moreover,

not only very generically formulated using the models' joint probabilities but it is also very straight-forward to apply.

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

APPENDIX

A: NOISY-OR

The truncated free energy takes on the following form for Noisy-OR:

$$\mathcal{F}_{NOR}(\mathcal{K}, \Theta) = N \sum_h \log (1 - \pi_h) + \sum_n \log \sum_{\vec{s} \in \mathcal{K}^{(n)}} \exp \widetilde{\mathcal{F}}$$

$$\widetilde{\mathcal{F}}(\vec{s}, \Theta) := \sum_h s_h \log \left( \frac{\pi_h}{1 - \pi_h} \right)$$

$$+ \sum_d y_d^n \log \left( \frac{1}{\prod_h (1 - W_{dh} s_h)} - 1 \right)$$

$$+ \sum_h \log(1 - W_{dh} s_h)$$

The M-step equations for noisy-OR are obtained by taking derivatives of the free energy, equating them to zero and solving the resulting set of equations. We report the results here for completeness:

$$\pi_h^{new} = \frac{1}{N} \sum_n \langle s_h \rangle_{q^n} \tag{11}$$

$$W_{dh}^{new} = 1 + \frac{\sum_n (y_d^n - 1) \langle D_{dh}(\vec{s}) \rangle_{q^n}}{\sum_n \langle C_{dh}(\vec{s}) \rangle_{q^n}} \tag{12}$$

where

$$D_{dh}(\vec{s}) := \frac{\widetilde{W}_{dh}(\vec{s}) s_h}{N_d(\vec{s})(1 - N_d(\vec{s}))}$$

$$C_{dh}(\vec{s}) := \widetilde{W}_{dh}(\vec{s}) D_{dh}(\vec{s}) \tag{13}$$

$$\widetilde{W}_{dh}(\vec{s}) := \prod_{h' \neq h} (1 - W_{dh'} s_{h'})$$

The update rule for $\vec{\pi}$ is quite straightforward. The update equations for the weights $W_{dh}$, on the other hand, do not allow a closed form solution (i.e. no exact M-step equation can be derived). The rule presented here, instead, expresses each $W_{dh}^{new}$ as a function of all current $\vec{W}$; this is a fixed-point equation whose fixed point would be the exact solution of the maximization step. Rather than solving the equation numerically at each step of the learning algorithm, we exploit the fact that in practice one single evaluation of 13 is enough to (noisily, not optimally) move towards convergence. Since TV-EM is guaranteed to never decrease $\mathcal{F}$, drops of the free-energy during training can only be ascribed to this fixed-point equation; this provides a simple mechanism to check and possibly correct for misbehaviors of 13 if needed.

B: M-STEP UPDATE RULES FOR BSC

The free energy for BSC follows from inserting (10) into (2). Update rules can be obtained by optimizing the resulting expression separately for the model parameters $\pi, \sigma^2$ and $W$ (compare, e.g., Henniges et al., 2010). For the sake of completeness, we show the result here:

$$\pi = \frac{1}{N} \sum_{n=1}^N \sum_{h=1}^H \langle s_h \rangle_{q^n} \tag{14}$$

$$\sigma^2 = \frac{1}{ND} \sum_{n=1}^N \left\langle || \vec{y}^{(n)} - W\vec{s} ||^2 \right\rangle_{q^n} \tag{15}$$

$$W = \left( \sum_{n=1}^{N} \vec{y}^{(n)} \langle \vec{s} \rangle_{q^n}^T \right) \left( \sum_{n'=1}^{N} \langle \vec{s}\vec{s}^T \rangle_{q^{n'}} \right)^{-1} \tag{16}$$

Exact EM can be obtained by setting $q^n$ to the exact posterior $p(\vec{s} \,|\, \vec{y}^{(n)}, \Theta)$. As this quickly becomes computational intractable with higher latent dimensionality, we approximate exact posteriors by truncated variational distributions (3). For BSC, the truncated free energy (4) takes the form

$$\mathcal{F}(\mathcal{K}, \Theta) = -\frac{ND}{2} \log\left(2\pi\sigma^2\right) + NH \log\left(1 - \pi\right) + \sum_n \log \left( \sum_{\vec{s} \in \mathcal{K}_n} \exp \left( \widetilde{\log p}(\vec{y}^{(n)}, \vec{s}|\Theta) \right) \right) \tag{17}$$

where

$$\widetilde{\log p}(\vec{y}, \vec{s}|\Theta) = -\frac{1}{2\sigma^2}(\vec{y} - W\vec{s})^T(\vec{y} - W\vec{s}) + |\vec{s}| \log\left(\frac{\pi}{1 - \pi}\right) \tag{18}$$

C: FURTHER EXPERIMENTAL RESULTS FOR BSC

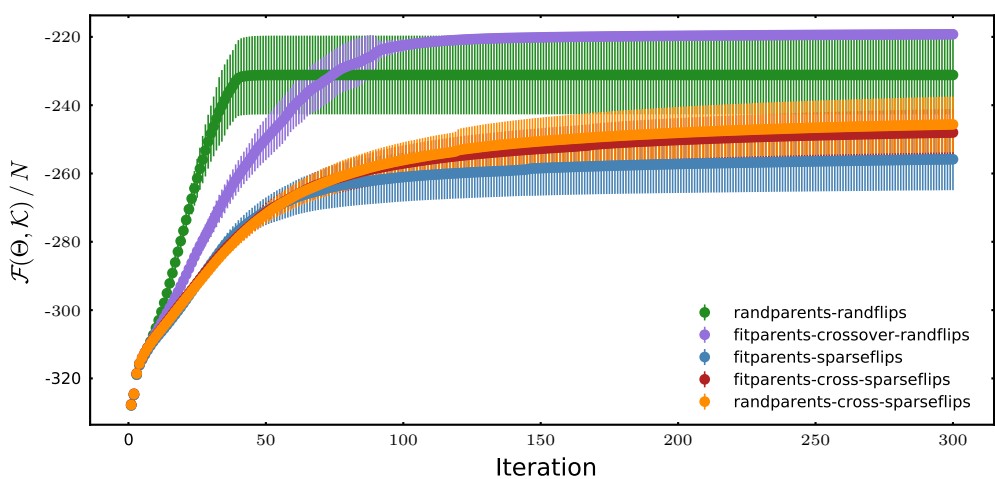

Figure 8: Results of the experiment with artificial data (10x10 bars) for the BSC model. Depicted is the evolution of the free energy for different EAs averaged over 20 independent runs. Dots and vertical errorbars show the mean and the standard deviation, respectively.

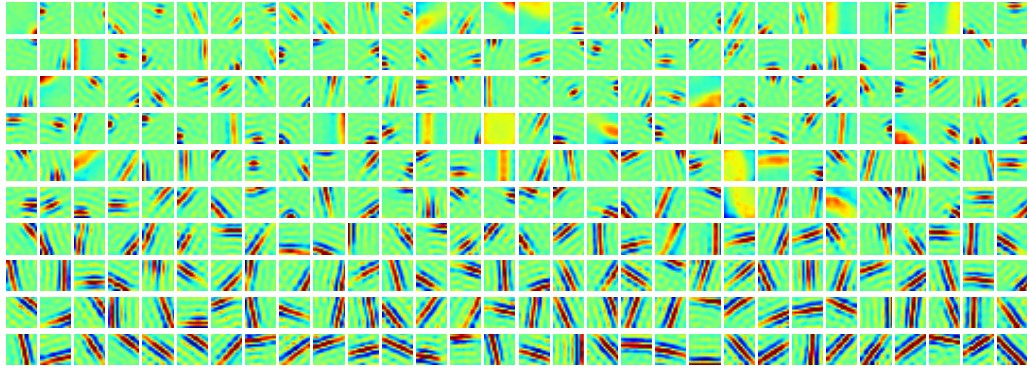

Figure 9: Full dictionary learned from natural images by the BSC model trained with the "fitparents-cross-sparseflips" EA. Depicted is the dictionary at iteration 4,000. The generative fields are ordered according to their activation, starting with most active fields.

D: SPARSITY-DRIVEN BITFLIPS

When performing sparsity-driven bitflips, we flip each bit of a particular child $\vec{s}^*$ with probability $p_0$ if it is 0, with probability $p_1$ otherwise. We call $p_{bf}$ the average probability of flipping any bit in $\vec{s}^*$. We impose the following constraints on $p_0$ and $p_1$:

- $p_1 = \alpha p_0$ for some constant $\alpha$
- the average number of on bits after mutation is set at $\widetilde{s}$

which yield the following expressions for $p_0$ and $p_1$:

$$\alpha = \frac{(H - |\vec{s}|) \cdot ((Hp_{bf}) - (\widetilde{s} - |\vec{s}|))}{(\widetilde{s} - |\vec{s}| + Hp_{bf})|\vec{s}|}$$

$$p_0 = \frac{Hp_{bf}}{H + (\alpha - 1)|\vec{s}|}$$

$$p_1 = \alpha \cdot p_0$$

Trivially, random uniform bitflips correspond to the case $p_0 = p_1 = p_{bf}$.

E: RELIABILITY OF EEM FOR NOISY-OR ON OVERLAPPING BARS

With respect to the tests shown in figure 4 and discussed in section 5.1, it is worth to spend a few more words on comparisons with the other algorithms shown (Lücke & Sahani, 2008, Fig. 9). Quantitative comparison to NMF approaches, neural nets (DI Spratling et al., 2009), and MCA (Lücke & Sahani, 2008) shows that EMM for noisy-OR performs well but there are also approaches with higher reliability. Of all the approaches which recover more than 15 bars on average, most require additional assumptions. E.g., all NMF approaches, non-negative sparse coding (Hoyer, 2004) and R-MCA$_2$ require constraints on weights and/or latent activations. Only MCA$_3$ does not require constraints and presumably neither DI. DI is a neural network approach, which makes the used assumptions difficult to infer. MCA$_3$ is a generative model with a max-non-linearity as superposition model. For learning it explores all sparse combinations with up to 3 components. Applied with $H = 32$ latents, it hence evaluates more than 60000 states per data point per iteration for learning. For comparison, EEM for noisy-OR evaluates on the order of $S = 100$ states per data point per iteration.

F: HIGHER-SCALE NATURAL IMAGE PATCHES FOR NOISY-OR

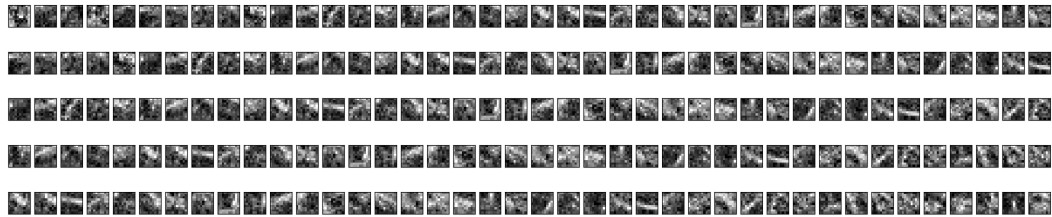

Figure 10: Generative fields learned running EEM for noisy-OR ("fitparents-sparseflips") for 175 iterations with $H = 200$ latent variables. Learned crowdedness $\pi H$ was 1.6.

