# OpenReview forum: "Evolutionary Expectation Maximization for Generative Models with Binary Latents"
_ICLR.cc/2018/Conference — Reject_

### Official Review · AnonReviewer1 · 2017-11-26
**Very specialised evolutionary EM algorithm**

**Rating:** 4
**Confidence:** 4

**Review:**

The paper presents a combination of evolutionary computation (EC) and variational EM for models with binary latent variables represented via a particle-based approximation.

The scope of the paper is quite narrow as the proposed method is only applicable to very specialised models. Furthermore, the authors do not seem to present any realistic modelling problems where the proposed approach would clearly advance the state of the art. There are no empirical comparisons with state of the art, only between different variants of the proposed method.

Because of these limitations, I do not think the paper can be considered for acceptance.

Detailed comments:

1. When revising the paper for next submission, please make the title more specific. Papers with very broad titles that only solve a very small part of the problem are very annoying.

2. Your use of crossover operators seems quite unimaginative. Genomes have a linear order but in the case of 2D images you use it is not obvious how that should be mapped to 1D. Combining crossovers in different representations or 2D crossovers might fit your problem much better.

3. Please present a real learning problem where your approach advances state of the art.

4. For the results in Fig. 7, please run the algorithm until convergence or justify why that is not necessary.

5. Please clarify the notation: what is the difference between y^n and y^(n)?

---

> ### Author Response · Authors · 2017-12-18
> **title should be amended, as suggested. replies to other comments**
>
> We agree that the title as-is (which only consists of the given name of the novel learning algorithm here developed) can be misleading in that this paper has smaller scope than a full replacement of standard variational EM techinques. It will therefore be amended to reflect the applicability of the method uniquely to models with binary latent variables.
>
> The crossover is performed on one-dimensional bit-strings (the latent states), so we are not sure a 2D crossover would benefit the algorithm.
>
> We posted another comment in reply to the other issues raised here.

---

### Official Review · AnonReviewer2 · 2017-11-27
**lacks comparisons to non-EA approaches, hard to gain insight on how to design the EA substep for new applications**

**Rating:** 4
**Confidence:** 4

**Review:**

## Review summary

Overall, the paper makes an interesting effort to tightly integrate
expectation-maximization (EM) training algorithms with evolutionary algorithms
(EA). However, I found the technical description lacking key details and the
experimental comparisons inadequate. There were no comparisons to non-
evolutionary EM algorithms, even though they exist for the models in question.
Furthermore, the suggested approach lacks a principled way to select
and tune key hyperparameters. I think the broad idea of using EA as a substep
within a monotonically improving free energy algorithm could be interesting,
but needs far more experimental justification.


## Pros / Stengths

+ effort to study more than one model family

+ maintaining monotonic improvement in free energy


## Cons / Limitations

- poor technical description and justification of the fitness function

- lack of comparisons to other, non-EA algorithms

- lack of study of hyperparameter sensitivity


## Paper summary

The paper suggests a variant of the EM algorithm for binary hidden variable
models, where the M-step proceeds as usual but the E-step is different in two
ways. First, following work by J. Lucke et al on Truncated Posteriors, the
true posterior over the much larger space of all possible bit vectors is
approximated by a more tractable small population of well-chosen bit vectors,
each with some posterior weight. Second, this set of bit vectors is updated
using an evolutionary/genetic algorithm. This EA is the core contribution,
since the work on Trucated Posteriors has appeared before in the literature.
The overall EM algorithm still maintains monotonic improvement of a free
energy objective.

Two well-known generative models are considered: Noisy-Or models for discrete
datasets and Binary Sparse Coding for continuous datasets. Each has a
previously known, closed-form M-step (given in supplement). The focus is on
the E-step: how to select the H-dimensional bit vector for each data point.

Experiments on artificial bars data and natural image patch datasets compare
several variants of the proposed method, while varying a few EA method
substeps such as selecting parents by fitness or randomly, including crossover
or not, or using generic or specialized mutation rates.


## Significance

Combining evolutionary algorithms (EA) within EM has been done previously, as
in Martinez and Vitria (Pattern Recog. Letters, 2000) or Pernkopf and
Bouchaffra (IEEE TPAMI, 2005) for mixture models. However, these efforts seem
to use EA in an "outer loop" to refine different runs of EM, while the present
approach uses EA in a substep of a single run of EM. I guess this is
technically different, but it is already well known that any E-step method
which monotonically improves the free energy is a valid algorithm. Thus, the
paper's significance hinges on demonstrating that the particular E step chosen
is better than alternatives. I don't think the paper succeeded very well at
this: there were no comparisons to non-EA algorithms, or to approaches that
use EA in the "outer loop" as above.


## Clarity of Technical Approach

What is \tilde{log P} in Eq. 7? This seems a fundamental expression. Its
plain-text definition is: "the logarithm of the joint probability where
summands that do not depend on the state s have been elided". To me, this
definition is not precise enough for me to reproduce confidently... is it just
log p(s_n, y_n | theta)? I suggest revisions include a clear mathematical
definition. This omission inhibits understanding of this paper's core
contributions.

Why does the fitness expression F defined in Eq. 7 satisfy the necessary
condition for fitness functions in Eq. 6? This choice of fitness function does
not seem intuitive to me. I think revisions are needed to *prove* this fitness
function obeys the comparison property in Eq. 6.

How can we compute the minimization substep in Eq. 7 (min_s \tilde{logP})? Is
this just done by exhaustive search over bit vectors? I think this needs
clarification.


## Quality of Experiments

The experiments are missing a crucial baseline: non-EA algorithms. Currently
only several varieties of EA are compared, so it is impossible to tell if the
suggested EA strategies even improve over non-EA baselines. As a specific
example, previous work already cited in this paper -- Henniges et al (2000) --
has developed a non-EA EM algorithm for Binary Sparse Coding, which already
uses the truncated posterior formulation. Why not compare to this?

The proposed algorithm has many hyperparameters, including number of
generations, number of parents, size of the latent space H, size of the
truncation, etc. The current paper offers little advice about selecting these
values intelligently, but presumably performance is quite sensitive to these
values. I'd like to see some more discussion of this and (ideally) more
experiments to help practitioners know which parameters matter most,
especially in the EA substep.

Runtime analysis is missing as well: Is runtime dominated by the EA step? How
does it compare to non-EA approaches? How big of datasets can the proposed
method scale to?

The reader walks away from the current toy bars experiment somewhat confused.
The Noisy-Or experiment did not favor crossover and and favored specialized
mutations, while the BSC experiment reached the opposite conclusions. How does
one design an EA for a new dataset, given this knowledge? Do we need to
exhaustively try all different EA substeps, or are there smarter lessons to
learn?



## Detailed comments

Bottom of page 1: I wouldn't say that "variational EM" is an approximation to
EM. Sometimes moving from EM to variational EM can mean we estimate posteriors
(not point estimates) for both local (example-specific) and global parameters.
Instead, the *approximation* comes simply from restricting the solution space
to gain tractability.

Sec. 2: Make clear earlier that hidden var "s" is assumed to be discrete, not
continuous.

After Mutation section: Remind readers that "N_g" is number of generations

---

> ### Author Response · Authors · 2017-12-18
> **clarification of the issues raised**
>
> Many thanks for the thorough review.
>
> Regarding hyperparameter sensitivity, our experience suggests that EEM is robust w.r.t. changes in all hyperparameters (within reasonable bounds), the most significant hyperparameter being the size of the sets of latent states K. Indeed, the paper would benefit from the addition of a systematic study of hyperparameter sensitivity. Future revisions will  add such a section (most likely to the appendix).
>
> Regarding the choice of fitness function, as discussed in the paper the only requirement is that it satisfies (6) (and, pragmatically, that it is cheap to compute). Any such fitness function would guarantee that adopting states with higher fitness would also increase the free energy (our true objective function).
> Nonetheless, we will add a more rigorous definition of \tilde{log P} in future revisions, as well as amend equation (7) -- the minimum should be taken over all states in the set K^n, not over all possible states. That second factor on the right hand side of (7) is only there to guarantee positivity of the fitness function (necessary for fitness-proportional parent selection).
>
> We also agree with the reviewer that a generic procedure to select the best EA for a given generative model is desirable. In fact, it would be best if this procedure was automatic and did not require user intervention. Work in this direction is underway.
> Improving our understanding of the difference in performance of the different EAs for different models will also be part of future work on this topic.
>
> We posted another comment in reply to the issues raised here that are common with the other reviewers.

---

> > ### Comment · AnonReviewer2 · 2018-01-12
> > **Response to Author Feedback and Other Reviews**
> >
> > Thanks to the authors for their reply. For this review cycle, I stand by my original rating of "4".  I think the broad idea of using EA as a substep within a monotonically improving free energy algorithm could be interesting, but needs far more experimental justification than presented here as well as more insightful suggestions about how to select the best EA procedure (use crossover? use mutations?) for new problems.
> >
> > I'm glad to hear that clarifications about hyperparameters and the definition of logP are on the TODO list. These are definitely needed to help others understand and deploy this method effectively.

---

### Official Review · AnonReviewer3 · 2017-11-28
**Interesting combination of evolutionary algorithms and EM; however, key technical details are missing and evaluation is contrived**

**Rating:** 4
**Confidence:** 4

**Review:**

This paper proposes an evolutionary algorithm for solving the variational E step in expectation-maximization algorithm for probabilistic models with binary latent variables. This is done by (i) considering the bit-vectors of the latent states as genomes of individuals, and by (ii) defining the fitness of the individuals as the log joint distribution of the parameters and the latent space.

Pros:
The paper is well written and the methodology presented is largely clear.

Cons:
While the reviewer is essentially fine with the idea of the method, the reviewer is much less convinced of the empirical study. There is no comparison with other methods such as Monte carlo sampling.
It is not clear how computationally Evolutionary EM performs comparing to Variational EM algorithm and there is neither experimental results nor analysis for the computational complexity of the proposed model.
The datasets used in the experiments are quite old. The reviewer is concerned that these datasets may not be representative of real problems.
The applicability of the method is quite limited. The proposed model is only applicable for the probabilistic models with binary latent variables, hence it cannot be applied to more realistic complex model with real-valued latent variables.

---

> ### Author Response · Authors · 2017-12-18
> **will amend the title to reflect applicability to latent variables only**
>
> We agree that the title as-is (which only consists of the given name of the novel learning algorithm here developed) can be misleading in that this paper has smaller scope than a full replacement of standard variational EM techinques. It will therefore be amended to reflect the applicability of the method uniquely to models with binary latent variables.
>
> We posted another comment in reply to the other issues raised here.

---

### Author Response · Authors · 2017-12-18
**regarding the absence of competitive benchmarks**

We thank the reviewers for their comments. As several of the points raised appear to be similar for all reviewers, we will address them here together. Other more specific points will be clarified in their relevant comment thread.
As underlined by abstract and discussion, we see the main contribution of this paper to be the development of a variational EM learning algorithm that directly employs evolutionary optimization techniques to monotonically increase a variational free-energy. Experimental results are provided as a proof of concept to show 1) general viability of the approach and 2) scalability up to hundreds of latent variables (also for models with challenging posterior structure such as noisy-OR).
The authors hope the theoretical link established here will pave the way to a wider range of new  techniques which can leverage results on both variational and evolutionary approaches. We stressed that the presented results are "a proof of concept" (see abstract), and they were as such not meant to compete with current benchmarks, nor did we focus our research on achieving such results at this stage. We are happy that all reviewers appreciated the general novel direction, i.e., novel combination of different research fields. At the same time, we are, of course, disappointed that the absence of numerical results that were competitive with recent benchmarks was rated so negatively.
We will follow the feedback of the reviewers to improve that shortcoming of the paper in future versions. Given the exceptional agreement of the reviewers on the current version, such efforts are presumably better targeted at a submission to another venue - and we thank the reviewers again for their feedback which, we believe, will improve such future submissions.

---

> ### Comment · AnonReviewer2 · 2018-01-12
> **RE: benchmarks**
>
> Just to clarify my position, I would suggest that any "proof of concept" paper DOES need to position itself clearly with respect to other approaches. You want your reader to walk away with clear understand of when you use your approach and why.
>
> Even a "conceptual" positioning would be a step in the right direction, such as clearly explaining an instance where your method would have lower runtime than an alternative E-step, or avoid local optima better than an alternative.
>
> This is not to say you need some exhaustive benchmark table. To me, it would be totally fine if you had a few results where your method was better, and some where it wasn't, as long as there was clear understanding of when your method improves on baselines and why.
>
> Good luck on future submissions!

---

### Decision · Program_Chairs · 2018-01-29
**ICLR 2018 Conference Acceptance Decision**

**Decision:**

Reject

**Comment:**

This method makes a connection between evolutionary and variational methods in a particular model.  This is a good contribution, but there has been little effort to position it in comparison to standard methods that do the same thing, showing relative strengths and weaknesses.

Also, please shorten the abstract.